# Microbial Characteristics of the Combined Ozone and Tea Polyphenols or Sodium Hypochlorite Disinfection in the Pipe Network

**Cuimin Feng** [1,2,*]**, Na Zhu** [1,2]**, Ying Li** [1,2]**, Zhen Xu** [1,2] **and Ziyu Guo** [3]

[1] Key Laboratory of Urban Stormwater System and Water Environment, Ministry of Education, Beijing University of Civil Engineering and Architecture, Beijing 100044, China; zhu_nana@126.com (N.Z.); 1109liying@163.com (Y.L.); xuzhen_0302@163.com (Z.X.)

[2] National Demonstration Center for Experimental Water Environment Education, Beijing University of Civil Engineering and Architecture, Beijing 100044, China

[3] Beijing General Municipal Engineering Design & Research Institute Co., Ltd., Beijing 100082, China; gu_oziyu@126.com

\* Correspondence: feng-cuimin@sohu.com

**Abstract:** Microbiological safety of water in the pipe network is an important guarantee for safe drinking water. Simulation tests of stainless steel pipe network were carried out using te4a polyphenols and sodium hypochlorite as auxiliary disinfectants for ozone disinfection to analyze the persistent disinfection effects of different combined disinfection methods by measuring the changes in *total bacterial colonies* in the water. High-throughput sequencing of microorganisms in the pipe network was performed to analyze the differences in the community structure of microorganisms in the water and pipe wall under different disinfection methods. The results showed that the application of auxiliary disinfectants had a relatively long-lasting inhibitory effect on the *bacterial colonies* in the water, and the diversity of microorganisms in the pipe network varied significantly. As an auxiliary disinfectant for ozone disinfection, tea polyphenols are more powerful than sodium hypochlorite in killing pathogens and chlorine-resistant bacteria, so they are more beneficial to ensure the microbiological safety of water in stainless steel pipe networks.

**Keywords:** drinking water; ozone; tea polyphenols; disinfection; microorganisms; pipe network

## 1. Introduction

The safety of drinking water is vital to human health, and disinfection is an important guarantee for safety. Chlorine disinfection is widely used in water plant practice due to its better bactericidal effect and sustained action [1], but at the same time, a large number of disinfection by-products have emerged, and more than 700 chlorine disinfection by-products have been detected, with different toxic effects posing a great threat to human health [2]. Ozone is highly oxidizing and has obvious disinfection advantages compared to chlorine disinfection, which can achieve effective killing of bacteria, viruses and protozoa at low concentrations within a short contact time [3,4]. Ozone can also reduce the organic content of water, provide odor and smell control, and oxidize heavy metals, etc. [5] However, ozone is more active and lacks the persistent disinfection ability, so it is not used for disinfection alone [5]. In order to avoid the shortcomings of single disinfection technology, two or more disinfection methods are usually used in combination [6,7]. Therefore, the combined application of auxiliary disinfectants after ozone disinfection is considered to improve the safety of drinking water.

Compared with liquid chlorine, sodium hypochlorite has the advantages of high safety, convenient use and easy storage [8]. Sodium hypochlorite forms hypochlorous

acid through hydrolysis, then further decomposes to form nascent oxygen [O], which has a strong oxidizing effect on microorganisms and achieves a killing effect [8]. Therefore, the combined ozone and sodium hypochlorite disinfection can be applied to extend the sterilization time and reduce the dosage of sodium hypochlorite [9]. Tea polyphenols are a class of polyphenols extracted from tea leaves with high content and variety, which have broad-spectrum resistance to a variety of bacteria, fungi and viruses, and their antibacterial effects are manifested in both inhibition and killing [10–12]. Due to the superior antibacterial properties and sustained action of tea polyphenols, they can be used as disinfectants to provide security for drinking water, and the research related to this has progressed. Xie et al. [13] showed that tea polyphenols can be used for disinfection of different water sources, which can achieve good disinfection effect. However, tea polyphenols are used as the only disinfectant with high dosage and poor economy, so it is recommended to be used as an auxiliary disinfectant in conjunction with the main disinfection process [14,15]. Feng et al. [16] used the response surface method to design the combined disinfection effect test of ozone and tea polyphenols, and the ideal working conditions were obtained by simulation.

The microbiological safety needs to be ensured in the drinking water distribution system, so there is a need for overall consideration of the water and pipe wall microorganisms as microorganisms on the pipe wall will enter the water under the effect of water flow or by aging off and become a source of bacteria in the water [17]. Therefore, the effect of disinfection methods from the perspective of microorganisms themselves need to be studied in depth. In this study, stainless steel pipe was chosen as the research object, which has good wear resistance, corrosion resistance, ductility and toughness, and is gradually being used in water supply networks [18,19]. The effects of ozone disinfection ($O_3$), the combined ozone/tea polyphenols disinfection ($O_3$/TP) and the combined ozone/sodium hypochlorite disinfection ($O_3$/NaClO) were analyzed by using a pipe network simulation system, to investigate the effects of different disinfection methods on the distribution characteristics of microorganisms in water and pipe walls. Then, the disinfection ability from the perspective of microorganisms was analyzed in order to clarify the feasibility of tea polyphenols as auxiliary disinfectants, providing a certain theoretical basis for the application of combined ozone/tea polyphenols disinfection technology.

## 2. Materials and Methods

### 2.1. Test Material

The National I test piece was selected as the simulated pipe wall hanging piece, made of stainless steel, with the specifications of 50.0 mm × 25.0 mm × 2.0 mm (surface area of 28 cm²) and surface roughness of Δ7.

The tea polyphenols were purchased from Beijing Zhongke Quality Control Biotechnology Co., Ltd., Beijing, China, and were in the form of yellow powder at room temperature with a purity of AR99.0%. The test results showed that the content of catechins accounted for 90%, and the content of Epigallocatechin gallate (EGCG) accounted for 56%.

The test raw water was taken from the city water supply network and left for 2–3 days to remove the residual chlorine, which was used to simulate the filtered water from the water plant. The water quality test results are as follows: pH 7.81, turbidity 0.96 NTU, chroma 1 degree (°), total bacteria 540 CFU·mL$^{-1}$ and residual chlorine 0.01 mg·L$^{-1}$.

### 2.2. Test Device and Operating Conditions

The dynamic simulation system of the pipe network is shown in Figure 1. The test adopted a non-circular operation mode to simulate the stainless steel water supply pipe network system, and explored the characteristics of microorganisms in water and pipe walls where ozone, the combined ozone/tea polyphenols method and the combined ozone/sodium hypochlorite method were applied as disinfectants. The core of this device is biological annular reactor (BAR) with an effective volume of 6 L, and 24 test pieces were

added to each reactor. During the test, the reactors were wrapped in tinfoil to avoid changes in water temperature caused by direct sunlight, and the water temperature was maintained at 22 °C by temperature sensors and temperature control rods.

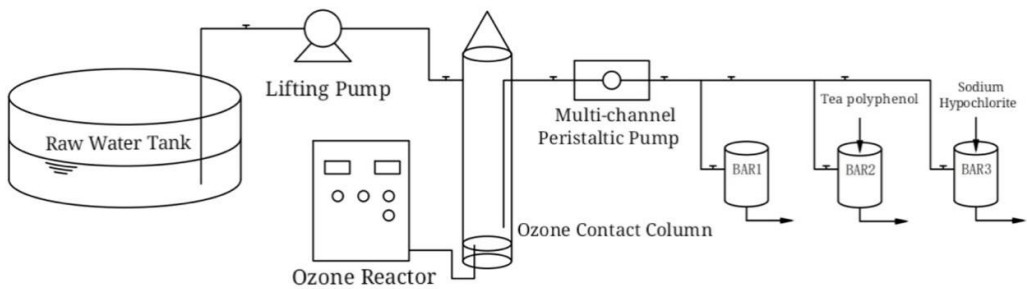

**Figure 1.** Schematic diagram of test equipment.

In the test operation, the raw water entered the ozone contact column through the lifting pump and after disinfection by ozone, the effluent entered BARs through the multi-channel peristaltic pump, then no auxiliary disinfectant was added to BAR1, tea polyphenols and sodium hypochlorite were added to BAR2 and BAR3, respectively. The disinfection effect was affected by the amount of ozone dosage, ozone contact time and auxiliary disinfectant dosage. Inlet flow of BARs were controlled as 0.6 L·h⁻¹ and simulated hydraulic retention time was 10 h during the test.

*2.3. Testing Indexes and Methods*

Ferrous tartrate-standard curve method was used to determine the concentration of tea polyphenols. First the regression equation of the standard curve was determined, and then the regression equation was used to calculate the concentration of tea polyphenols in the water during the test. Iodometry method was used to determine the concentration of ozone in the water, and starch solution was used as an indicator.

The total number of bacteria in the water was determined according to the *Standard Examination Methods for Drinking Water* (GB/T 5750.12-2006) by the plate count method [20]. The test run time was for a total of 35 days, every 5 days for each reactor in the same location of the simulated pipe wall hanging piece of colony count test.

For microorganisms in the water, 6 L of water samples were taken from each reactor on the 35th day of the dynamic test. Then, filter membranes with a pore size of 0.22 μm were selected and the water samples were filtered with a vacuum extraction bottle to enrich the amount of bacteria and extract DNA. After passing the test, the V3–V4 region of 16S rRNA was amplified by PCR, and high-throughput sequencing was performed after successful amplification. Microbial community analysis was performed according to the sequencing results.

For microorganisms on the pipe wall, 10 mock pipe wall hangings were taken from each reactor on the 35th day of the dynamic test. First, they were placed in a beaker containing 300 mL of sterile water, and then placed in ultrasonic cleaning shaking. The same filter membranes with a pore size of 0.22 μm were selected, and then the above procedure was repeated.

**3. Results and Discussion**

*3.1. Analysis of Disinfection Effect*

A stainless steel network simulation system was set up to analyze the persistent disinfection effect of different disinfection methods by detecting the total amount of bacteria in the effluent of the three BARs. Ozone and tea polyphenols were dosed with reference

to the results obtained in the preliminary work of the research group: ozone dosage of 2.5 mg·L⁻¹, ozone contact time of 25 min, and tea polyphenols dosage of 20 mg·L⁻¹ [16]. Sodium hypochlorite was dosed at 2 mg·L⁻¹ with reference to the operating experience of the water plant during the test [21,22]. The total bacterial colonies in the water under different disinfection methods were measured at intervals of 5 days, as shown in Figure 2.

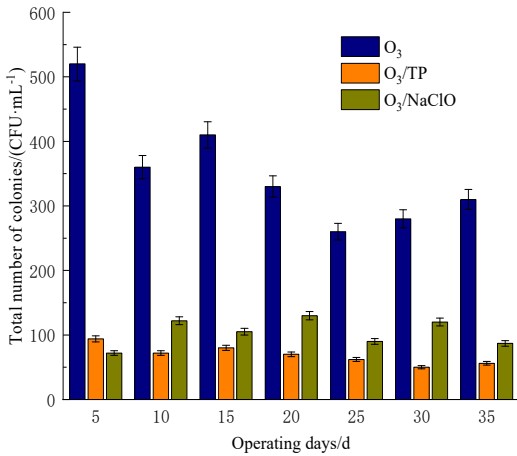

**Figure 2.** The total bacterial colonies in the water under different disinfection methods.

It can be seen that the total bacterial colonies in the water was less than 100 CFU·mL⁻¹ within 5 days after the combined ozone and tea polyphenols or sodium hypochlorite disinfection (Figure 2), which meets the requirements of the Standards for Drinking Water Quality (GB5749-2006) [20]. After that, until 35 days, the total bacterial colonies in the water with tea polyphenols as auxiliary disinfectant were always maintained below 100 CFU·mL⁻¹ and continued to show a decreasing trend. However, there was a rebound phenomenon with sodium hypochlorite as an auxiliary disinfectant within 10–35 days, and the water quality did not meet the standard requirements. It shows that the combined ozone/tea polyphenols disinfection has a relatively long-lasting inhibitory effect and the water quality of the stainless steel pipe network is safer among the three disinfection methods. The reason may be that during the disinfection process, the action of tea polyphenols is stable and lasts for a long time, while sodium hypochlorite has a rapid effect and decays quickly at the same time. Hence, the synergistic effect of sodium hypochlorite and ozone has a better inhibitory effect on microorganisms in the early stage. However, the attenuation of tea polyphenols is slower, so its continuous action makes the control effect on bacteria stronger. In addition, the oxidizing free radicals generated in the water after ozone dosing will cause tea polyphenols to exhibit pro-oxidation at low concentrations, increasing the amount of ·OH in the water, and enhancing the sterilization effect of O₃/TP combined disinfection.

### 3.2. Microbial Diversity Analysis

High-throughput sequencing of the bacteria in the water and the pipe wall samples at the end of the dynamic operation of the three BARs, then the OTU at 97% similarity level were subjected to bioinformatic statistical analysis after de-redundancy to reflect the abundance and diversity of the microbial communities by the Alpha diversity index (Table 1).

**Table 1.** The Alpha diversity index (water/pipe wall).

| Disinfection Method | Coverage | Shannon | ACE | Chao1 | Simpson |
|---|---|---|---|---|---|
| $O_3$ | 1.00/1.00 | 0.78/1.57 | 256.18/410.12 | 229.30/368.33 | 0.69/0.41 |
| $O_3$/TP | 1.00/1.00 | 1.89/2.50 | 258.46/353.86 | 243.64/319.12 | 0.34/0.14 |
| $O_3$/NaClO | 1.00/0.95 | 5.72/3.22 | 1407.10/474.57 | 1403.53/413.18 | 0.01/0.22 |

ACE index and Chao1 index are often used in ecology to estimate the total number of species, with larger indices indicating higher microbial richness. Shannon index and Simpson index are often used in ecology to describe biodiversity, with larger Shannon and smaller Simpson indices indicating higher community diversity [23]. It can be concluded that ACE, Chao1 and Shannon increased and Simpson decreased in the water when the auxiliary disinfectants were applied in combination with ozone compared to ozone disinfection from Table 1. This indicates that the microbial species diversity and richness in the water increased with combined disinfection compared to ozone alone. Among them, ACE and Chao1 increased slightly and Simpson decreased from 0.69 to 0.34 when tea polyphenols was used as auxiliary disinfectant, while ACE and Chao1 increased significantly and Simpson decreased from 0.69 to 0.01 when sodium hypochlorite was used as auxiliary disinfectant. Combined with Figure 2, it can be seen that the combined $O_3$/TP disinfection has better bacterial inhibition effect and lower diversity of microorganisms in the water compared with the combined $O_3$/NaClO disinfection.

Microbial diversity analysis of the pipe wall found ACE and Chao1 were 353.86 and 319.12 respectively when the combined disinfection of $O_3$/TP was used, both of which decreased compared to ozone disinfection. This indicates that the microbial richness of the pipe wall was reduced, which may be due to the effect of metal ions in stainless steel on the disinfection of tea polyphenols. Studies have shown that the phenolic hydroxyl groups in the structure of tea polyphenols can complex with most metal ions (such as $Cd^{2+}$, $Ca^{2+}$, $Cu^{2+}$, $Fe^{3+}$, etc.), forming tea polyphenols-metal complexes with some differences in bacterial inhibitory activity [24]. In contrast, the Alpha diversity indices for the combined $O_3$/NaClO disinfection reflected an increase in the microbial diversity and richness on the pipe wall compared to the ozone disinfection. This indicates that different auxiliary disinfectants have different effects on the biofilm of the pipe wall. Therefore, it is necessary to analyze the microbial characteristics of the water and the pipe wall to investigate the differences in the effectiveness of different combined disinfection from the bacteria themselves.

### 3.3. Analysis of Microbial Characteristics in the Water

The combined $O_3$/TP disinfection has the bacterial inhibition advantage over the combined $O_3$/NaClO disinfection, and the microorganisms in the water have a lower diversity. However, the specific characteristics of bacterial community structure need to be further analyzed.

### 3.3.1. Microbial Diversity Analysis Based on Phylum Level

The raw water was separately disinfected by $O_3$, $O_3$/TP and $O_3$/NaClO, then stayed in the stainless steel pipe network dynamically for 10 h. After that, high-throughput sequencing was performed on microorganisms in the water, and the colony structure at phylum level is shown in Figure 3. The bacteria were mainly *Proteobacteria* in the water after ozone disinfection, whose content accounted for 97.8%, followed by *Euryarchaeota* accounted for 2.1%. *Proteobacteria* are all Gram-negative bacteria, containing a variety of metabolic species, both aerobic and anaerobes, both autotrophic and heterotrophic, and they are common bacteria in drinking water pipe networks [25]. *Proteobacteria* are currently the largest phylum within the bacteria domain, including several pathogens such as *Escherichia coli, Salmonella, Vibrio, Pseudomonas aeruginosa*, etc. [26] As a large branch of

archaea, *Euryarchaeota* contain most species such as *Methanogenus* and *Halobacterium*. The microbial community structure in the water became simpler when the combined disinfection of O₃/TP was used, with *Proteobacteria* reaching 99.95% and *Euryarchaeota* not exceeding 0.01%. In contrast, the diversity of bacteria increased significantly after the combined O₃/NaClO disinfection. In addition to 48.62% of *Proteobacteria*, there were *Firmicutes* (19.69%), *Bacteroidetes* (9.38%), *Actinobacteria* (4.14%), *Nitrospirae* (3.26%) and *Euryarchaeota* (0.61%), etc. The unicity of the colony structure in the water after the combined O₃/TP disinfection shows that this method has a more obvious inhibitory effect on *Euryarchaeota*, *Firmicutes*, *Bacteroides*, etc., than the combined O₃/NaClO disinfection.

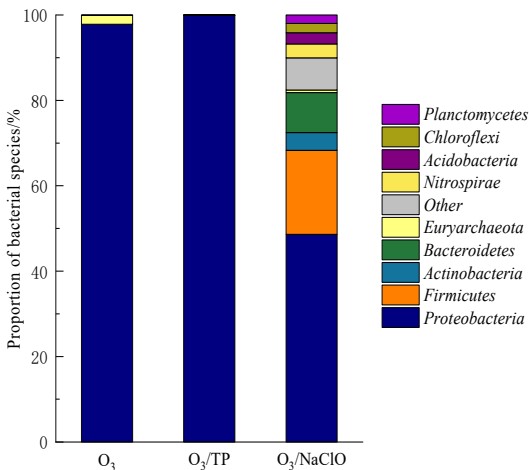

**Figure 3.** Colony structure at phylum level.

### 3.3.2. Microbial Diversity Analysis Based on Class Level

There is no obvious difference in the characteristics of the bacterial community between ozone and the combined O₃/TP disinfection at phylum level, further analysis is needed at class level (Figure 4). It can be seen that the dominant flora of *Proteobacteria* after the three disinfection methods was different at class level. The dominant flora is *Gammaproteobacteria* with 82.92%, followed by *Alphaproteobacteria* with 14.67% after ozone disinfection. When tea polyphenols were used as an auxiliary disinfectant, *Alphaproteobacteria* is predominant with 74.86%, followed by *Betaproteobacteria* (18.51%) and *Gammaproteobacteria* (6.58%). This indicates that tea polyphenols are more effective in killing *Gammaproteobacteria*. The species and genetic diversity of *Proteobacteria* is extremely rich [26], among them, *Alphaproteobacteria* include plant symbiotic bacteria, animal symbiotic bacteria, and are widely present in the water environment. *Betaproteobacteria* include many aerobic bacteria or facultative bacteria, which can be detected in environmental samples such as wastewater or soil. *Gammaproteobacteria* include some taxa that are important in medicine and scientific research, such as *Enterobacteraceae*, *Vibrionaceae* and *Pseudomonadaceae*, etc. [26]. Microorganisms in the water were more complex after the combined O₃/NaClO disinfection, with *Betaproteobacteria*, *Gammaproteobacteria* and *Alphaproteobacteria* accounting for 21.94%, 15.99% and 9.22%, respectively. Besides, *Bacilli*, *Clostridia*, *Actinobacteria*, *Bacteroidia* and *Sphingobacteriia* also had high percentages of 9.68%, 7.86%, 4.14%, 3.88% and 3.75%, respectively. *Bacilli* and *Clostridia* belong to *Firmicutes*, and Gram staining is positive [27]. Many *Firmicutes* can produce budding spores to resist dehydration and extreme environments. *Actinobacteria* are also a group of Gram-positive bacteria [27], about 80% of widely used antibiotics are produced by various *Actinobacteria*, such as streptomycin, oxytetracycline, tetracycline, gentamicin, etc. [28]. *Bacteroidia* and *Sphingobacteriia* belong to *Bacteroidetes*, and many bacteria of *Bacteroidia* live in the intestinal tract of humans or animals, and in some cases, become pathogens. The test results show that these groups of

bacteria are tolerant to chlorine but sensitive to tea polyphenols, which are related to the characteristics of cell membranes of different classes of bacteria [29]. This indicates that the growth of many common microorganisms are better controlled during the combined O₃/TP disinfection compared to the combined O₃/NaClO disinfection, and has better inhibitory effects on chlorine-resistant groups of *Bacilli, Actinobacteria, and Sphingobacteriia,* etc.

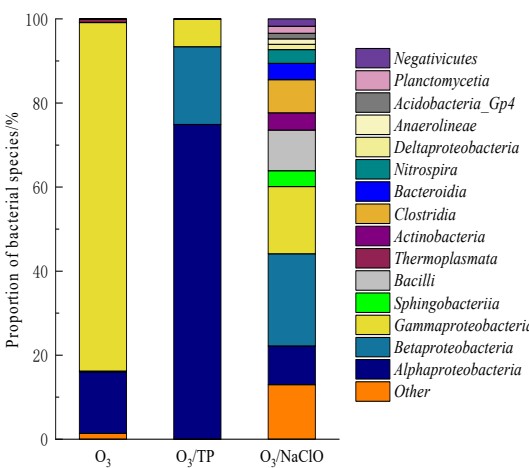

**Figure 4.** Colony structure at class level.

### 3.3.3. Microbial Diversity Analysis Based on Genus Level

The differences in the flora in the pipe network water are very obvious at genus level, as shown in Figure 5. The microbial species were relatively simple when ozone disinfection was used, with *Pseudomonas* as the dominant genus accounting for about 80%, followed by *Blastomonas* at about 10%. While the percentages of *Pseudomonas* and *Blastomonas* detected were significantly reduced after the combined *disinfection,* indicating that the auxiliary disinfectants can effectively control the growth of these two types of bacteria. *Blastomonas* is Gram-negative bacteria that reproduce by germination, *Pseudomonas* is widely present in water, soil and atmosphere by flagellar movement, and Gram stain is also negative, among which *P. aeruginosa, P. pseudomallei* and *P. fluorescens* are all opportunistic pathogens present in the environment [30]. As a representative of this genus, *P. aeruginosa* has several pathogenic factors, and the presence of it was detected by Anversa et al. [31] during an investigation of the quality of public water supplies in municipalities in São Paulo State, Brazil, indicating the ability of this bacterium to resist conventional water treatment. Huang et al. [32] found that the sources of contamination were diversified through the analysis of the contamination of *P. aeruginosa* in water sources, drinking water, and pipeline water supply systems. Since *Pseudomonas* has certain resistance to UV, ozone and chlorine disinfection, its proportion was relatively high during ozone disinfection in the experiment, and the percentage decreased significantly after the addition of auxiliary disinfectants. Yi et al. [33] showed that tea polyphenols have a significant effect on the membrane protein of *P. aeruginosa*, promoting the disorder of bacterial metabolism and gradually destroying the cell structure, which play a role in inhibitory. After sodium hypochlorite is added, it is hydrolyzed to produce hypochlorous acid, which acts on microorganisms through various mechanisms to achieve disinfection [8]. *Alcaligenes* and *Rhizobacter*, which have slightly different flagella morphological characteristics from *Pseudomonas*, were detected with sodium hypochlorite and tea polyphenols as auxiliary disinfectants, respectively, accounting for a relatively small percentage.

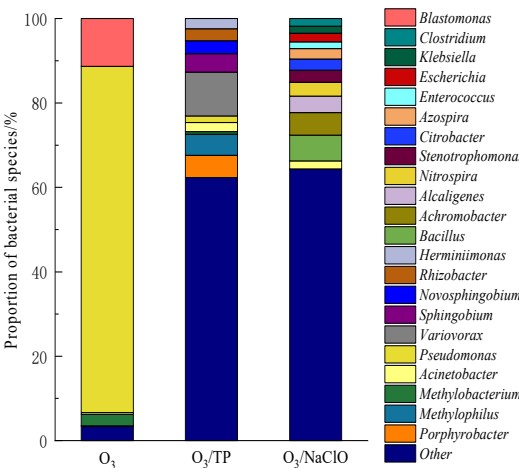

**Figure 5.** Colony structure at genus level.

It can be seen that the microbial community structure within the water samples was more complex during combined disinfection from Figure 5. *Variovorax*, *Porphyrobacter*, *Methylophilus*, and *Sphingobium* accounted for 10.39%, 5.24%, 5.02% and 4.33%, respectively, when the combined O₃/TP *disinfection was used*. While *Bacillus*, *Achromobacter*, *Stenotrophomonas*, *Citrobacter*, *Escherichia* and *Klebsiella* accounted for a higher percentage with 6.09%, 5.34%, 2.87%, 2.65%, 2.05% and 1.65%, respectively, when the combined O₃/NaClO *disinfection was used*. Among them, *Citrobacter*, *Escherichia* and *Klebsiella* belong to *Enterobacteriaceae* and are used as a microbiological indicator of water quality and fecal contamination. Additionally, they have varying degrees of drug resistance, as opportunistic pathogens have a greater impact on human health [34]. In contrast, these *Enterobacteriaceae* microorganisms accounted for a smaller proportion after the combined O₃/TP *disinfection*, indicating that tea polyphenols are more effective than sodium hypochlorite as an auxiliary disinfectant in controlling the above-mentioned opportunistic pathogens, and the biosafety of the water was higher. Liu et al. [35] found that UV/tea polyphenols combined disinfection was significantly more effective than UV/sodium hypochlorite combined disinfection in killing most of the common environmental microbial populations, and also had a strong ability to kill pathogens and chlorine-resistant bacteria, which is similar to the results of this study. That is, the use of tea polyphenols as an auxiliary disinfectant in combination with the main disinfection process is more secure for controlling water microbial safety.

### 3.4. Analysis of Microbial Characteristics of the Pipe Wall

Microorganisms will use the water body organic matter for metabolism, growth and reproduction, and they attach to the pipe wall to form biofilm during the operation of the pipe network. Many bacteria will grow and reproduce in the microbial community on the pipe wall and enter the water under the action of water flushing or aging off [36]. From the overall safety of the pipe network, the community structure of the pipe wall microorganisms is analyzed.

### 3.4.1. Microbial Diversity Analysis Based on Phylum Level

The colony structure of the pipe wall at phylum level derived from high-throughput sequencing is shown in Figure 6, which is similar to the distribution of microorganisms in the water. The microorganisms were all dominated by *Proteobacteria*, accounting for 97.91%, 98.82%, and 87.17%, respectively, after the disinfection of O₃, O₃/TP and O₃/NaClO. While the percentage of *Firmicutes*, *Bacteroidetes* and *Actinobacteria* were also higher after the combined O₃/NaClO disinfection, with 4.62%, 3.72%, and 0.94%,

respectively. This is similar to the finding of Sun et al. [25] that the larger population on biofilm is *Proteobacteria*, followed by *Firmicutes*, *Actinobacteria*, etc., in the drinking water distribution system. That is, these bacteria are widely present in the water supply system, adaptable and widespread. Studies have also shown that the main factor affecting the structure of the biofilm community on the inner wall of the pipeline at phylum level is the pipe material [36], follow-up on this point of view needs to be analyzed through comparative experiments.

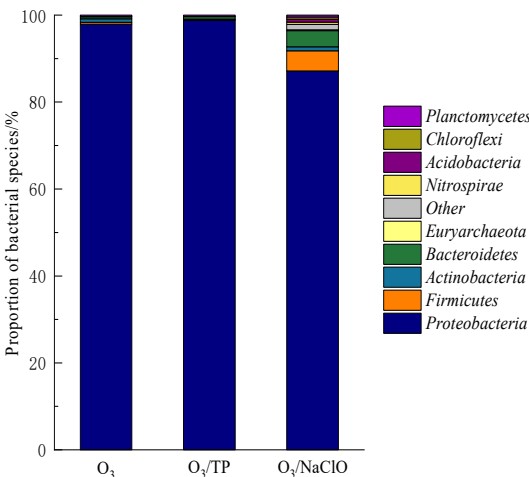

**Figure 6.** Colony structure at phylum level.

### 3.4.2. Microbial Diversity Analysis Based on Class Level

The colony structure at class level is shown in Figure 7. *Gammaproteobacteria* was the dominant bacteria after ozone disinfection, accounting for 61.27%, followed by *Alphaproteobacteria* (29.94%) and *Betaproteobacteria* (6.68%). When the combined $O_3$/TP disinfection was used, *Alphaproteobacteria* was the dominant bacteria with 61.07%, followed by *Gammaproteobacteria* (21.44%) and *Betaproteobacteria* (16.29%). When the combined $O_3$/NaClO disinfection was used, the percentage of *Proteobacteria* was 87.11% on the pipe wall, much greater than 47.15% in the water, with the specific distribution of *Betaproteobacteria* (68.92%), *Gammaproteobacteria* (11.51%) and *Alphaproteobacteria* (6.68%). In addition, *Clostridia*, *Sphingobacteriia*, *Bacilli* and *Actinobacteria* also had high percentages with 2.57%, 2.19%, 1.74%, and 0.94%, respectively. The above data indicate that tea polyphenols as an auxiliary disinfectant has a stronger inhibitory effect on *Betaproteobacteria* compared to sodium hypochlorite. Li et al. [37] showed that *Betaproteobacteria* and *Gammaproteobacteria* were more resistant to chlorine than *Alphaproteobacteria* in biofilms, which is similar to the results of $O_3$/NaClO disinfection in this experiment. *Betaproteobacteria* are the most widely distributed class in nature and are widely found in lakes, rivers and drinking water biofilms [38,39]. It has been shown that *Betaproteobacteria* have the ability to secrete Extracellular Polymeric Substances (EPS), which make microorganisms adhere to the inner surface of pipes, enhance the hydrophobicity of cells, and aggregate into clusters with other microorganisms [40]. Under the protection of EPS, microorganisms on biofilm are protected from the direct action of disinfection and have access to more nutrients. Liu et al. [41] studied the effects of tea polyphenols on the quorum sensing and virulence factors of *Klebsiella Pneumoniae* and found that tea polyphenols inhibited population sensing activity and reduced their motility, protease and extracellular polysaccharide production, then inhibited biofilm formation. Huber et al. [42] also reported that plant polyphenols could interfere with bacterial population sensing and inhibit bacterial biofilm formation. Liu et al. [35] found that tea polyphenols probably destroyed tyrosine-like proteins and degraded humic acids, thereby affecting the growth of pipe wall microorganisms by

analyzing the fluorescence properties of organic matter in pipe wall biofilm when UV/tea polyphenols combined disinfection. Moreover, catechins and epicatechins were detected in the biofilm of the pipe network, which indicated that tea polyphenols would interact with the pipe wall microorganisms [35]. Therefore, the characteristics and environmental factors of the tea polyphenols disinfection pipe wall microorganisms in pipe networks still need to be studied in depth with a focus on the dual role and conditions of tea polyphenols on the inhibition and assimilation of pipe wall microorganisms to clarify the microbial safety of tea polyphenol disinfection of pipe wall networks.

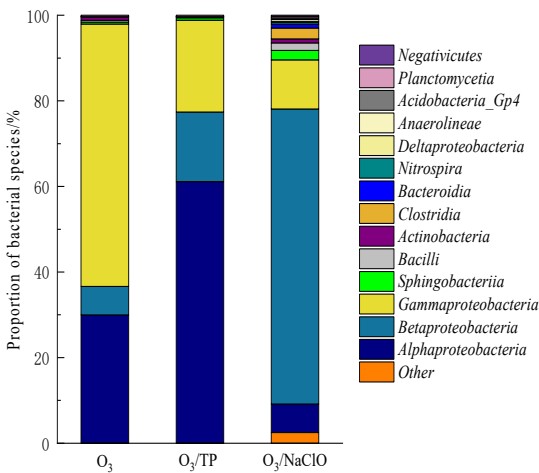

**Figure 7.** Colony structure at class level.

### 3.4.3. Microbial Diversity Analysis Based on Genus Level

The colony structure at genus level is shown in Figure 8. It can be seen that the distribution of microorganisms was relatively simple after ozone disinfection, with *Pseudomonas* dominating about 60.67%, followed by *Blastomonas* (13.69%), *Methylobacterium* (13.11%), and *Curvibacter* (3.86%). *Pseudomonas* were found to exhibit excellent biofilm formation in a variety of environments, and they provide protection against chemical and mechanical attack [43]. In contrast, the percentage of *Pseudomonas* were significantly reduced and the dominant genus changed when ozone and auxiliary disinfectants were combined for disinfection. The microorganisms with a higher percentage were *Methylobacterium* (23.62%), *Rhizobacter* (15.13%), *Methylophilus* (6.17%), *Pseudomonas* (6%) and *Curvibacter* (3.91%) after the combined O$_3$/TP disinfection. The dominant bacterium *Methylobacterium* belongs to *Methylobacteriaceae* and *Alphaproteobacteria*. Tsagkar et al. [44] found that it is a key strain in the formation of bacterial aggregates in water and subsequent surface biofilm formation, which may account for its high percentage in the pipe wall flora. The percentage of *Curvibacter* increased significantly to about 46.87% after the combined O$_3$/NaClO disinfection, and this bacterium belongs to *Comamonas*, *Betaproteobacteria*, a new genus identified by Ding and Yokota of the University of Tokyo in 2004 [45]. In addition, there were *Pseudomonas* (6.71%), *Duganella* (4.87%), *Achromobacter* (2.19%), *Novosphingobium* (1.84%), *Alcaligenes* (1.46%) and *Bacillus* (1.39%) under this method. That is, the pipe wall microorganisms are more complex when sodium hypochlorite is used as an auxiliary disinfectant. The distribution of microorganisms in the water and the pipe wall has some correlation and difference, which needs to be compared and analyzed.

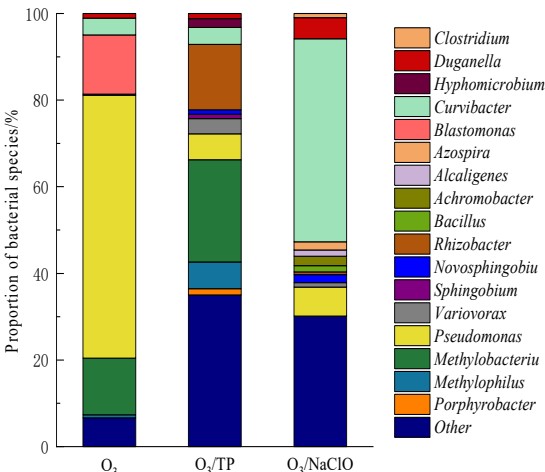

**Figure 8.** Colony structure at genus level.

*3.5. Comparative Analysis of Microorganisms in the Water and the Pipe Wall*

From the analysis in 2.3 and 2.4, it is clear that the bacterial community composition is significantly different under the action of different disinfection methods, which has a greater correlation with the nature of the bacteria themselves. In this study, the analysis of biodiversity can clarify the differences in bacterial tolerance to the environment under the three disinfection methods. The differences in bacterial community structure at class level are presented as an example (Table 2).

**Table 2.** Percentage of microorganisms at class level *(water/pipe wall)*.

| Disinfection method / Bacterial composition | O₃ (%) | O₃/TP (%) | O₃/NaClO (%) |
|---|---|---|---|
| *Alphaproteobacteria* | 14.67/29.94 | 74.86/61.07 | 9.22/6.68 |
| *Betaproteobacteria* | 0.21/6.68 | 18.51/16.29 | 21.94/68.92 |
| *Gammaproteobacteria* | 82.92/61.27 | 6.58/21.44 | 15.99/11.51 |
| *Sphingobacteriia* | -/0.47 | 0.01/0.57 | 3.75/2.19 |
| *Bacilli* | 0.02/0.42 | 0.02/0.09 | 9.68/1.74 |
| *Clostridia* | -/0.03 | -/0.02 | 7.86/2.57 |
| *Bacteroidia* | -/- | -/- | 3.88/0.87 |
| *Nitrospira* | -/- | -/- | 3.26/0.52 |
| *Planctomycetia* | -/0.33 | -/0.30 | 1.66/0.35 |
| *Other* | 1.36/0.04 | 0.02/0.04 | 12.98/2.49 |

Note: "-" means the bacterium is not detected or the percentage of detection is relatively small, which is classified as other.

It can be seen from Table 2 that the type of disinfectant affects the composition and structure of the bacterial community and the dominant bacteria detect vary, however, the dominant bacteria are consistent in the water and the pipe wall for the same disinfection method. In general, ozone disinfection can effectively remove *Betaproteobacteria*, and O₃/TP disinfection has a strong inhibitory effect on *Gammaproteobacteria*. The two also have a better removal effect on *Bacilli*, *Clostridia*, *Bacteroidia*, etc., which were poorly inhibited when O₃/NaClO disinfection was used. However, O₃/NaClO disinfection can better remove *Alphaproteobacteria* than that of O₃/TP disinfection. Studies have shown that tea polyphenols have antibacterial activity against both Gram-negative and Gram-positive bacteria, but have a stronger effect on the latter [46]. Most of *Alphaproteobacteria* are Gram-negative bacteria, so it accounts for a higher proportion when O₃/TP disinfection was used. Douterelo et al. [47] measured significantly different bacterial community

composition in biofilms and water samples in an experimental drinking water distribution system and found that *Betaproteobacteria* and *Gammaproteobacteria* were the dominant bacteria in the biofilm in the pipeline, while *Alphaproteobacteria* dominated in the water samples. It is similar to these test results, but Douterelo et al. chose only sodium hypochlorite as disinfectant during their study. Geng et al. [48] found that *Bacillus subtilis* spores are resistant to chlorine disinfection alone, but the inactivation effects of them are greatly enhanced when ozone/chlorine combined disinfection has a synergistic effect. Tachikawa et al. [49] found that the sequential disinfection of ozone and hydrogen peroxide showed a good synergistic effect, and the removal effect of biofilm was better. Combined disinfection has certain application prospects, and its mechanism still needs to be explored in depth. For the development of biological safety control measures for drinking water, the criteria measured include total bacteria, opportunistic pathogens or pathogens, changes in biodiversity, biofilm and bioerosion, etc. [50]. Therefore, the analysis of microbial diversity in the pipe network is relevant for the selection of auxiliary disinfectants. It also provides scientific support for the application of ozone/tea polyphenols combined disinfection systems.

## 4. Conclusions

(1) The *total bacterial colonies* in the water were well controlled when the auxiliary disinfectant was combined with ozone for disinfection in stainless steel pipe network simulation system. When using tea polyphenols and sodium hypochlorite as auxiliary disinfectants, the *total bacterial colonies* in the water remained below 100 CFU·mL$^{-1}$ for 35 days and 5 days, respectively. This indicates that the combined ozone/tea polyphenols disinfection is more sustainable and the water quality in the pipe network is safer.

(2) Microbial species diversity and richness in the water increased when combined with auxiliary disinfectants compared to ozone disinfection. The combined ozone/tea polyphenols disinfection has better bacterial inhibition effect and lower microbial diversity in the water compared with the combined ozone/sodium hypochlorite disinfection. Microbial diversity increased and richness decreased on the pipe wall when the combined ozone/tea polyphenols disinfection was used, this may be due to the effect of metal ions in the stainless steel pipe on the disinfection of tea polyphenols. However, microbial diversity and richness on the pipe wall both increased when the combined ozone/sodium hypochlorite disinfection was used. This indicates that different auxiliary disinfectants combined with ozone have different effects on the biofilm organisms on the pipe wall.

(3) For microorganisms in the water, the dominant bacteria were approximately the same at phylum and class levels before and after the application of auxiliary disinfectants. However, the proportions were different and the other bacterial groups varied considerably. Compared with sodium hypochlorite as an auxiliary disinfectant, the ozone/tea polyphenols combined disinfection was better at removing common microorganisms and chlorine-resistant bacteria such as *Bacilli*, *Actinobacteria* and *Sphingobacteriia*, etc. At genus level, the microbial species were simple after ozone disinfection, *Pseudomonas* and *Blastomonas* were the dominant genera with a total share of about 90%, while the killing effect on both was significant after combined disinfection. The combined ozone/tea polyphenol disinfection was

better in killing *Citrobacter*, *Escherichia*, and other opportunistic pathogens, which improved the biosafety compared to sodium hypochlorite as an auxiliary disinfectant.

(4) For microorganisms of the pipe wall, *Proteobacteria* were the main phylum under the three disinfection methods. The dominant genera at class level were *Gammaproteobacteria*, *Alphaproteobacteria*, *Betaproteobacteria*, respectively, when ozone, the combined ozone/tea polyphenols disinfection and the combined ozone/sodium hypochlorite were applied as disinfectants, which were related to the difference in the effects of different disinfectants on bacteria. At genus level, the microbial distribution was highly variable, with *Pseudomonas*, *Methylobacterium* and *Curvibacter*, etc. all showing some biofilm formation ability. The distribution of microorganisms in water and pipe wall had some correlation and difference, and the correlation showed that the dominant genera in water and pipe wall were unified by different disinfection methods. The difference was manifested in the different growth ability of the same microorganism in water and pipe wall, for example, *Betaproteobacteria* tend to form biofilms on the inner wall of the pipe, so their proportion in the wall microorganisms is significantly larger than that in water. Microorganisms on the pipe wall will enter the water under the effect of water flow or aging off in actual operation and become the source of bacteria in the water. Therefore, it is necessary to take measures to suppress the microorganisms in the pipe network to ensure the safety of drinking water.

(5) Combined disinfection has more advantages in microbiological safety control than ozone alone disinfection. In the stainless steel pipe network, the composition of bacteria in the water is complex and the microbial risk is high after the combined ozone/sodium hypochlorite disinfection. *It has* a persistent *disinfection effect* and small microbial diversity in the water when tea polyphenols are used as auxiliary disinfectants, besides, the inhibitory effect on chlorine-resistant bacteria and opportunistic pathogens in the water is stronger than that of sodium hypochlorite. Therefore, the combined ozone/tea polyphenol disinfection method can better ensure the safety of drinking water in the pipe network.

**Author Contributions:** Conceptualization, C.F., N.Z. and Z.G.; methodology, Z.G. and N.Z.; software, Z.X. and N.Z.; validation, C.F., N.Z. and Y.L.; formal analysis, N.Z. and Z.X.; resources, C.F.; data curation, Z.G. and N.Z.; writing—original draft preparation, N.Z.; writing—review and editing, C.F., Y.L. and N.Z.; visualization, Z.X.; supervision, C.F.; project administration, C.F.; funding acquisition, C.F. All authors have read and agreed to the published version of the manuscript.

**Funding:** This research was funded by the National Natural Science Foundation of China (51678026), Beijing University of Civil Engineering Postgraduate Innovation Project (PG2021047).

**Institutional Review Board Statement:** Not applicable.

**Informed Consent Statement:** Not applicable.

**Data Availability Statement:** The data that support the findings of this study are available from the corresponding author upon reasonable request.

**Conflicts of Interest:** The authors declare no conflict of interest.

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
