# Peer review of "Microbial Characteristics of the Combined Ozone and Tea Polyphenols or Sodium Hypochlorite Disinfection in the Pipe Network"

_water, doi:10.3390/w13131835_

Round 1
Reviewer 1 Report
In this manuscript, Cuimin Feng et al studied the ozone disinfection in stainless steel pipe network using tea polyphenols or sodium hypochlorite as auxiliary disinfectants. The microbial species diversity and richness in water and on pipe wall were investigated. As an auxiliary disinfectant for ozone disinfection, tea polyphenols are found more powerful than sodium hypochlorite in killing pathogens and chlorine-resistant bacteria.
The current version of manuscript can be considered to publish with minor revision:
1. In the introduction section, the authors need to properly clarify if there are any similar studies regarding water disinfection by ozone and tea polyphenol. For example: Feng et al “ Disinfection Effects and Operating Conditions of Tea Polyphenols Combined with Ozone” Ozone: Science & Engineering, 2020, 42, 6
2. The tea polyphenols might be a mixture of several compounds. It will be the useful for the authors to provide more information about the possible products in the tea polyphenols they used.
Author Response
Point 1: In the introduction section, the authors need to properly clarify if there are any similar studies regarding water disinfection by ozone and tea polyphenol. For example: Feng et al “Disinfection Effects and Operating Conditions of Tea Polyphenols Combined with Ozone” Ozone: Science & Engineering, 2020, 42, 6 

Response 1: Thank you for your advice, the content of this part has been added to the introduction in the revision manuscript.
Point 2: The tea polyphenols might be a mixture of several compounds. It will be the useful for the authors to provide more information about the possible products in the tea polyphenols they used.
Response 2: Thank you for your advice, and more information has been added to 2.1 in the revision manuscript. Tea polyphenols are a kind of mixture, mainly composed of catechins, anthocyanins, phenolic acids and depside derivatives, among them, catechins are the main component of tea polyphenols. EGCG is the catechin with the highest content and strongest activity, which is also the main component of tea polyphenols with antibacterial effect. The catechin content of the tea polyphenols used in the experiment accounted for 90%, and the EGCG content accounted for 56%.
Reviewer 2 Report
This paper deals with tea poly-13 phenols and sodium hypochlorite as auxiliary disinfectants for ozone disinfection by simulation tests of stainless steel pipe network. The microorganisms were removed from water processes were investigated by several characterization techniques. The studies were quite systematic and the resulted were well organized by the authors. I’d like to recommend the publication of this paper in water after revision.
- Author shall provide whether there are any pollutants exited in the water after treatment.
- Author shall explain the mechanism of water treatment by O3, O3/TP and O3/NaClO.
- Author shall explain whether the element of stainless steel pipe network release in the water.
- According to Table 2, author shall explain which microorganisms can be eliminated effectively by O3, O3/TP and O3/NaClO.
- The study should explain what make tea polyphenols inhibit bacterial colonies superior than sodium hypochlorite.
- The study should provide RAMAN and FTIR result to show functional group of tea polyphenol.
